# Graph Stochastic Neural Networks for Semi-supervised Learning

**Haibo Wang**[1, 2,*] **Chuan Zhou**[3, 4] **Xin Chen**[2,*] **Jia Wu**[5] **Shirui Pan**[6] **Jilong Wang**[2]

[1]Department of Computer Science and Technology, Tsinghua University, Beijing, China
[2]Institute for Network Sciences and Cyberspace, Tsinghua University, Beijing, China
[3]Academy of Mathematics and Systems Science, Chinese Academy of Sciences, Beijing, China
[4]School of Cyber Security, University of Chinese Academy of Sciences, Beijing, China
[5]Faculty of Science and Engineering, Macquarie University, Sydney, Australia
[6]Faculty of Information Technology, Monash University, Melbourne, Australia
{wang-hb15, cx18}@mails.tsinghua.edu.cn, zhouchuan@amss.ac.cn,
jia.wu@mq.edu.au, shirui.pan@monash.edu, wjl@cernet.edu.cn

## Abstract

Graph Neural Networks (GNNs) have achieved remarkable performance in the task of the semi-supervised node classification. However, most existing models learn a deterministic classification function, which lack sufficient flexibility to explore better choices in the presence of kinds of imperfect observed data such as the scarce labeled nodes and noisy graph structure. To improve the rigidness and inflexibility of deterministic classification functions, this paper proposes a novel framework named **G**raph **S**tochastic **N**eural **N**etworks (GSNN), which aims to model the uncertainty of the classification function by simultaneously learning a family of functions, *i.e.*, a stochastic function. Specifically, we introduce a learnable graph neural network coupled with a high-dimensional latent variable to model the distribution of the classification function, and further adopt the amortised variational inference to approximate the intractable joint posterior for missing labels and the latent variable. By maximizing the lower-bound of the likelihood for observed node labels, the instantiated models can be trained in an end-to-end manner effectively. Extensive experiments on three real-world datasets show that GSNN achieves substantial performance gain in different scenarios compared with state-of-the-art baselines.

## 1 Introduction

Graphs are essential tools to represent complex relationships among entities in various domains, such as social networks, citation networks, biological networks and physical networks. Analyzing graph data has become one of the most important topics in the machine learning community. As an abstraction of many graph mining tasks, semi-supervised node classification, which aims to predict the labels of unlabeled nodes given the graph structure, node features and labels of partial nodes, has received significant attention in recent years. Graph Neural Networks (GNNs), in particular, have achieved impressive performance in the graph-based semi-supervised learning task [1, 2, 3, 4, 5].

Most existing GNN models are designed to learn a deterministic classification function. This kind of design makes them look simple and artistic, but the other side of the coin is that the deterministic classification function makes these GNN models lack sufficient flexibility to cater for kinds of imperfect observed data. For example, in many real situations, the ground-truth labels of nodes

---

are often expensive or difficult to obtain, which leads to the sparseness of the labeled nodes. The insufficient supervision information can easily lead to the overfitting of the deterministic classification functions, especially when there are no additional labeled nodes as the validation set for early-stopping. Another example is the noise in the graph structure. Arisen in nature or injected deliberately by attackers, noise is prone to affect the neighbor information aggregation and misleads the learning of deterministic classification functions. The rigidness and inflexibility of deterministic classification functions make it difficult for them to bypass these similar issues and explore better choices.

In line of the aforementioned observations, this paper proposes a novel **G**raph **S**tochastic **N**eural **N**etwork (GSNN for short) to model the uncertainty of GNN classification functions. GSNN aims to learn simultaneously a family of classification functions rather than fitting a deterministic function. This empowers GNNs the flexibility to handle the imperfection or noise in graph data, and further bypass the traps caused by sparse labeled data and unreliable graph structure in real applications.

Specifically, we treat the classification function to be learned as a stochastic function and integrate it into the process of label inference. To model the distribution of the stochastic function, we introduce a learnable neural network, which is coupled with a high-dimensional latent variable and takes the message-passing form. To infer the missing labels, we need to obtain the joint posterior distribution of labels for unlabeled nodes and the classification function, of which the exact form is intractable in general. To solve the problem, we adopt the amortised variational inference [6, 7] to approximate the intractable posterior distribution with the other two types of neural networks. By maximizing the lower-bound of the likelihood for observed node labels, we could optimize all parameters effectively in an end-to-end manner. We conduct extensive experiments on three real-world datasets. The results show that compared with state-of-the-art baselines, GSNN not only achieves comparable or better performance in the standard experimental scenario with early-stopping, but also shows substantial performance gain when labeled nodes are scarce (no early-stopping) and there are deliberate edge perturbations in the graph structure.

## 2    Related Work

**Graph Neural Networks for Graph-based Semi-supervised Learning:**  Recently, GNNs have been attracting considerable attention [8, 9, 10]. The early ideas are to derive different forms of the graph convolution in the spectral domain based on the graph spectral theory [1, 11, 2, 3, 12]. Bruna *et al.* [1] propose the first generation spectral-based GNN. To reduce the computational complexity, Defferrard *et al.* [2] propose to use a K-order Chebyshev polynomial to approximate the convolutional filter, which avoids intense calculations of eigendecomposition of the normalized graph Laplacian. Kipf and Welling [3] further simplify the graph convolution by the first-order approximation. which reduces the number of parameters and improves the performance. Another line of research is to directly perform graph convolution in the spatial domain [13, 4, 14, 15, 16]. Gilmer *et al.* [13] generalize spatial-based methods as a message-passing mechanism. Hamilton *et al.* [4] propose a general inductive framework, which could learn an embedding function that generalizes to unseen nodes. Veličković *et al.* [5] further introduce the attention mechanism, which assigns different weights to neighbor nodes and aggregate features with discrimination. Besides, other works also demonstrate that considering edge attributes [17], adding jumping connections [18] and modeling the outcome dependency [19] would be beneficial. However, these models generally learn a deterministic classification function, which lack sufficient flexibility to handle imperfect observed data such as the scarce labeled nodes and noisy graph structure.

**Uncertainty Modeling for Graph-based Semi-supervised Learning:** There are also some works using uncertainty modeling for graph-based semi-supervised learning, which are related to this paper [20, 21, 22]. Ng *et al.* [22] introduce Gaussian processes to model the semi-supervised learning problem on graphs, which mitigates the over-fitting to some extent. Zhang *et al.* [21] treat the observed graph as a realization from a parametric family of random graphs and propose bayesian graph convolutional neural networks to incorporate the uncertain graph information. Ma *et al* [20] further propose a flexible generative framework to model the joint distribution of the graph structure and the node labels. Most of these works typically model the uncertainty of the observed data (*e.g.*, graph structure). Different from them, in this paper, we view the classification function as a stochastic function and straightly model its distribution, which brings better performance in many scenarios.

# 3 Our Solution

In this paper, we define an undirected graph as $G = (V, E)$, where $V = \{v_1, ..., v_N\}$ represents a set of $N$ nodes and $E \subseteq V \times V$ is the set of edges. Let $\boldsymbol{A} \in \{0, 1\}^{N \times N}$ denote the binary adjacency matrix, $i.e.$, $\boldsymbol{A}_{u,v} = 1$ if and only if $(u, v) \in E$. Let $\boldsymbol{X} \in \mathbb{R}^{N \times F}$ be the node attribute matrix, where $F$ is the feature dimension and the feature vector of node $v$ is expressed as $\boldsymbol{x}_v$. Each node is labeled with one class in $C = \{c_1, ..., c_{|C|}\}$. In practice, only partial nodes come with labels. The set of these labeled nodes is denoted as $V_L$ and the set of unlabeled nodes is denoted as $V_U := V \setminus V_L$. For the task of semi-supervised node classification, given $\boldsymbol{A}$, $\boldsymbol{X}$ and the label information of $V_L$, the goal is to infer the labels of nodes in $V_U$ by learning a classification function $f$. The classification results can be denoted as $Y := \{\boldsymbol{y}_{v_1}, ..., \boldsymbol{y}_{v_N}\}$ where each $\boldsymbol{y}_.$ is a $|C|$-dimension probability distribution on $C$.

Most existing GNN models typically aim to learn a deterministic classification function, which lack sufficient flexibility to cater for kinds of imperfect observed data. For example, they are easy to overfit or be misled when labeled nodes are scarce or there exists noise in the graph structure. Therefore, instead of fitting a deterministic function, we here aim to learn a family of classification functions, which can be organized as a stochastic function $\mathfrak{F}$ with the distribution denoted as $p(f)$. Under this setting, the distribution of $Y$ can be formalized as follows:

$$p(Y|\boldsymbol{A}, \boldsymbol{X}) \triangleq \int p(f) p\big(Y|f(\boldsymbol{A}, \boldsymbol{X})\big) \, df = \int p(f) \prod_{v \in V} p\big(\boldsymbol{y}_v|f(\boldsymbol{A}, \boldsymbol{X})\big) \, df \tag{1}$$

where we use $p\big(Y|f(\boldsymbol{A}, \boldsymbol{X})\big)$ to denote the distribution of $Y$ corresponding to the classification function $f$. Eq. (1) assumes that the label inference for each node is conditionally independent, given a selected classification function $f$, the adjacency matrix $\boldsymbol{A}$ and the attribute matrix $\boldsymbol{X}$.

## 3.1 Framework for GSNN

In order to model the uncertainty of the classification function in Eq. (1), we here approximate the stochastic function $\mathfrak{F}$ using a learnable function $g_\varphi$ ($e.g.$, a neural network with parameters $\varphi$) with a random latent vector $\mathfrak{z}$ involved as below:

$$\mathfrak{F}(\boldsymbol{A}, \boldsymbol{X}) \triangleq g_\varphi(\boldsymbol{A}, \boldsymbol{X}; \mathfrak{z}) \tag{2}$$

where the prior distribution of $\mathfrak{z}$ is $p(\boldsymbol{z})$ defined as multivariate standard normal, $i.e.$, $p(\boldsymbol{z}) = \mathcal{N}(\boldsymbol{z}; \boldsymbol{0}, \boldsymbol{I})$. Note that the randomness of $\mathfrak{F}$ is induced by $\mathfrak{z}$ and the expression capacity of $\mathfrak{F}$ is captured by the structure of $g_\varphi(.;.)$. Combined Eq. (2) with Eq. (1), the distribution $p(Y|\boldsymbol{A}, \boldsymbol{X})$ can be rewritten as follows:

$$p(Y|\boldsymbol{A}, \boldsymbol{X}) = \int p(\boldsymbol{z}) \prod_{v \in V} p\big(\boldsymbol{y}_v|g_\varphi(\boldsymbol{A}, \boldsymbol{X}; \boldsymbol{z})\big) \, d\boldsymbol{z} \tag{3}$$

where $p\big(\boldsymbol{y}_v|g_\varphi(\boldsymbol{A}, \boldsymbol{X}; \boldsymbol{z})\big)$ is a distribution on $C$ of node $v$. In the semi-supervised transductive setting, the label information for labeled nodes in $V_L$ is also known. Denote $Y_L := \{\boldsymbol{y}_v\}_{v \in V_L}$ and $Y_U := Y \setminus Y_L$. Under the above setting, the conditional distribution of $Y_U$, given $\boldsymbol{A}$, $\boldsymbol{X}$ and $Y_L$, can be formalized as follows:

$$p(Y_U|\boldsymbol{A}, \boldsymbol{X}, Y_L) \triangleq \int p(\boldsymbol{z}|\boldsymbol{A}, \boldsymbol{X}, Y_L) \prod_{v \in V_U} p\big(\boldsymbol{y}_v|g_{\varphi_{Y_L}}(\boldsymbol{A}, \boldsymbol{X}; \boldsymbol{z})\big) \, d\boldsymbol{z} \tag{4}$$

where $p(\boldsymbol{z}|\boldsymbol{A}, \boldsymbol{X}, Y_L)$ is the posterior distribution of the latent vector $\mathfrak{z}$ and $\varphi_{Y_L}$ are parameters to be learned when $Y_L$ is taken into consideration.

We assume that the distributions $p(\boldsymbol{z}|\boldsymbol{A}, \boldsymbol{X}, Y_L)$ and $p\big(\boldsymbol{y}_v|g_{\varphi_{Y_L}}(\boldsymbol{A}, \boldsymbol{X}; \boldsymbol{z})\big)$ in Eq. (4) can be modeled by parametric families of distributions $p_\theta(\boldsymbol{z}|\boldsymbol{A}, \boldsymbol{X}, Y_L)$ and $p_\theta(\boldsymbol{y}_v|\boldsymbol{A}, \boldsymbol{X}, \boldsymbol{z})$ respectively, whose probability density function is differentiable almost everywhere $w.r.t.$ $\theta$. To predict $Y_U$ via modeling the distribution of the classification function, we need to obtain an *intractable* joint posterior $p_\theta(Y_U, \boldsymbol{z}|\boldsymbol{A}, \boldsymbol{X}, Y_L)$. To solve the problem, we adopt the variational inference. We introduce a variational distribution $q_\phi(Y_U, \boldsymbol{z}|\boldsymbol{A}, \boldsymbol{X}, Y_L)$ parameterized by $\phi$ to approximate the true posterior $p_\theta(Y_U, \boldsymbol{z}|\boldsymbol{A}, \boldsymbol{X}, Y_L)$. To learn the model parameters $\phi$ and $\theta$, we aim to optimize the evidence lower bound (ELBO) of the log-likelihood function for the observed node labels, $i.e.$, $\log p_\theta(Y_L|\boldsymbol{A}, \boldsymbol{X})$.

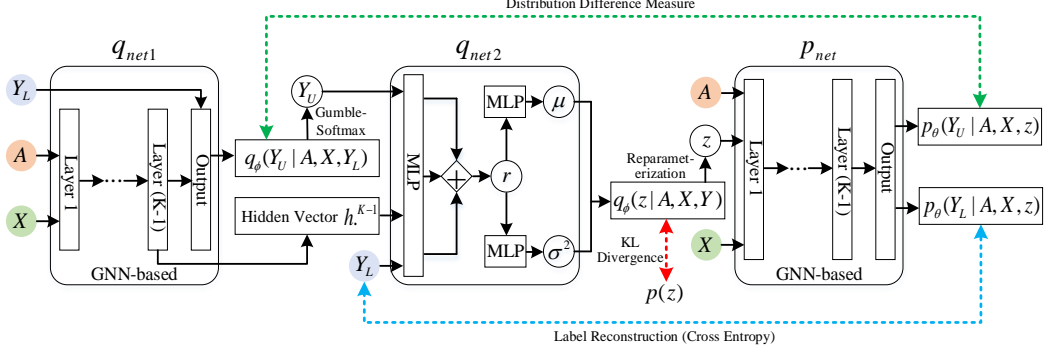

Figure 1: The overview of GSNN framework. $q_\phi(Y_U|\boldsymbol{A}, \boldsymbol{X}, Y_L)$, $q_\phi(\boldsymbol{z}|\boldsymbol{A}, \boldsymbol{X}, Y)$ and $p_\theta(Y|\boldsymbol{A}, \boldsymbol{X}, \boldsymbol{z})$ are respectively instantiated by $q_{net1}$, $q_{net2}$ and $p_{net}$. Colored circles represent the observed data, including $\boldsymbol{A}$, $\boldsymbol{X}$ and $Y_L$. $Y_U$ and $\boldsymbol{z}$ in the colorless circles are sampled from corresponding distributions. The blue, red and green dotted lines correspond to the first, second and third items of the ELBO objective function in Eq. (6) respectively.

Following the standard derivation of the variational inference, the ELBO objective function can be obtained as follows:

$$\log p_\theta(Y_L|\boldsymbol{A}, \boldsymbol{X}) \geq \mathbb{E}_{q_\phi(Y_U, \boldsymbol{z}|\boldsymbol{A}, \boldsymbol{X}, Y_L)} \left( \log p_\theta(Y|\boldsymbol{A}, \boldsymbol{X}, \boldsymbol{z}) + \log \frac{p(\boldsymbol{z})}{q_\phi(Y_U, \boldsymbol{z}|\boldsymbol{A}, \boldsymbol{X}, Y_L)} \right) \quad (5)$$
$$\triangleq \mathcal{L}_{ELBO}(\theta, \phi)$$

The variational joint posterior could be further factorized as $q_\phi(Y_U, \boldsymbol{z}|\boldsymbol{A}, \boldsymbol{X}, Y_L) = q_\phi(Y_U|\boldsymbol{A}, \boldsymbol{X}, Y_L) q_\phi(\boldsymbol{z}|\boldsymbol{A}, \boldsymbol{X}, Y)$ noting that $Y = Y_L \cup Y_U$. From a sampling perspective, it can be explained that the distribution of the random latent vector $\mathfrak{z}$ depends on the observed data and $Y_U$ sampled from the approximate posterior distribution $q_\phi(Y_U|\boldsymbol{A}, \boldsymbol{X}, Y_L)$. On this basis, the ELBO objective function can be rewritten as follows:

$$\mathcal{L}_{ELBO}(\theta, \phi) = \mathbb{E}_{q_\phi(Y_U|\boldsymbol{A}, \boldsymbol{X}, Y_L)} \mathbb{E}_{q_\phi(\boldsymbol{z}|\boldsymbol{A}, \boldsymbol{X}, Y)} \log p_\theta(Y_L|\boldsymbol{A}, \boldsymbol{X}, \boldsymbol{z}) -$$
$$\mathbb{E}_{q_\phi(Y_U|\boldsymbol{A}, \boldsymbol{X}, Y_L)} \mathrm{KL}\big(q_\phi(\boldsymbol{z}|\boldsymbol{A}, \boldsymbol{X}, Y) \,\|\, p(\boldsymbol{z})\big) - \quad (6)$$
$$\mathbb{E}_{q_\phi(Y_U|\boldsymbol{A}, \boldsymbol{X}, Y_L)} \big( \log q_\phi(Y_U|\boldsymbol{A}, \boldsymbol{X}, Y_L) - \mathbb{E}_{q_\phi(\boldsymbol{z}|\boldsymbol{A}, \boldsymbol{X}, Y)} \log p_\theta(Y_U|\boldsymbol{A}, \boldsymbol{X}, \boldsymbol{z})\big)$$

where $\mathrm{KL}(.\|.)$ represents the Kullback-Leibler divergence between two distributions. The first term of Eq. (6) is the opposite of the cross-entropy between ground-truth label vectors and the predicted class distributions for labeled nodes. The form of the third term is similar to the KL divergence, which characterizes the distribution difference between $q_\phi(Y_U|\boldsymbol{A}, \boldsymbol{X}, Y_L)$ and $p_\theta(Y_U|\boldsymbol{A}, \boldsymbol{X}, \boldsymbol{z})$. Based on the amortised variational inference [6, 7], $q_\phi(Y_U|\boldsymbol{A}, \boldsymbol{X}, Y_L)$, $q_\phi(\boldsymbol{z}|\boldsymbol{A}, \boldsymbol{X}, Y)$ and $p_\theta(Y|\boldsymbol{A}, \boldsymbol{X}, \boldsymbol{z})$ can be fitted by different types of neural networks, which would be described in detail in Section 3.2. The overall framework is referred as graph stochastic neural networks (GSNN for short), whose overview is shown in Fig. 1.

## 3.2 Model Instantiation, Training and Inference

In this part, we instantiate $q_\phi(Y_U|\boldsymbol{A}, \boldsymbol{X}, Y_L)$, $q_\phi(\boldsymbol{z}|\boldsymbol{A}, \boldsymbol{X}, Y)$ and $p_\theta(Y|\boldsymbol{A}, \boldsymbol{X}, \boldsymbol{z})$ with three neural networks (*i.e.*, $q_{net1}$, $q_{net2}$ and $p_{net}$) respectively.

**Instantiating $q_\phi(Y_U|\boldsymbol{A}, \boldsymbol{X}, Y_L)$ with $q_{net1}$:** The neural network $q_{net1}$ is designed into the form of message-passing. It consists of $K$ layers to aggregate the features of neighbor nodes with the following layer-wise propagation rule:

$$\boldsymbol{h}_v^k = \rho^{k-1}\Big( \sum_{u \in Ne\{v\} \cup \{v\}} a_{v,u}^{k-1} \boldsymbol{h}_u^{k-1} W_{q_{net1}}^{k-1} \Big), \quad k = 1, ..., K$$
$$q_\phi(\boldsymbol{y}_v|\boldsymbol{A}, \boldsymbol{X}, Y_L) = Cat(\boldsymbol{y}_v|\boldsymbol{h}_v^K), \quad v \in V_U \quad (7)$$

where $Ne\{v\}$ is the set of neighbor nodes of node $v$. $\boldsymbol{h}_v^k$ is the hidden representation for node $v$ in the $k^{th}$ layer and $\boldsymbol{h}_v^0 = \boldsymbol{x}_v$. The parameter $a_{v,u}^{k-1}$ represents the aggregation coefficient between

node $v$ and node $u$. The parameter matrix $W_{q_{net1}}^{k-1}$ represent the trainable parameters in the $k^{th}$ layer. The activation functions of the first $K-1$ layers (i.e., $\rho^0, ..., \rho^{K-2}$) are $ReLU$, and the activation function for the $K^{th}$ layer is $softmax$, which constructs the categorical distribution $Cat(.)$, i.e., $q_\phi(Y_U|\boldsymbol{A}, \boldsymbol{X}, Y_L)$. Note that $Y_L$ is not used as the input for $q_{net1}$, but as the supervision information for training $q_{net1}$ in the Eq. (10) below.

**Instantiating $q_\phi(\boldsymbol{z}|\boldsymbol{A}, \boldsymbol{X}, \boldsymbol{Y})$ with $\boldsymbol{q_{net2}}$:** The posterior distribution $q_\phi(\boldsymbol{z}|\boldsymbol{A}, \boldsymbol{X}, Y)$ depends on four parts of information: $\boldsymbol{A}$, $\boldsymbol{X}$, $Y_U$ and $Y_L$. Since $\boldsymbol{q_{net1}}$ has involved $\boldsymbol{A}$ and $\boldsymbol{X}$, we therefore directly use the hidden representations of the $(K-1)^{th}$ layer in $q_{net1}$ to represent $\boldsymbol{A}$ and $\boldsymbol{X}$. The unlabeled information $Y_U$ could be obtained by sampling from the output of $q_{net1}$ and $Y_L$ is directly taken as one of the input for $q_{net2}$. Inspired by variational auto-encoders [7], we let the variational posterior be a multivariate Gaussian with a diagonal covariance structure, which is flexible and could make the second item of the ELBO objection in Eq. (6) be computed analytically. Accordingly, $q_{net2}$ is designed as follows:

$$\boldsymbol{r}_v = \text{MLP}([\boldsymbol{h}_v^{K-1}||\boldsymbol{y}_v]), \quad v \in V$$
$$\boldsymbol{r} = \text{Readout}(\{\boldsymbol{r}_v\}_{v\in V}) \tag{8}$$
$$q_\phi(\boldsymbol{z}|\boldsymbol{A}, \boldsymbol{X}, Y) = \mathcal{N}\big(\boldsymbol{z}; \mu(\boldsymbol{r}), \sigma^2(\boldsymbol{r})\boldsymbol{I}\big)$$

where $.||.$ is the concatenation operation, MLP represents the multi-layer perceptron, Readout(.) function summarizes all input vectors into a global vector, and the MLP functions $\mu(.)$ and $\sigma^2(.)$ convert $\boldsymbol{r}$ into the mean and standard deviation, which parameterise the distribution of $q_\phi(\boldsymbol{z}|\boldsymbol{A}, \boldsymbol{X}, Y)$.

**Instantiating $p_\theta(\boldsymbol{Y}|\boldsymbol{A}, \boldsymbol{X}, \boldsymbol{z})$ with $\boldsymbol{p_{net}}$:** Given $\boldsymbol{z}$ sampled from $q_\phi(\boldsymbol{z}|\boldsymbol{A}, \boldsymbol{X}, Y)$, $p_{net}$ specifies an instance of the stochastic function $\mathfrak{F}$ (i.e., function $g$ defined in Eq. (2)). The network architecture of $p_{net}$ is similar to that of $q_{net1}$, which takes the sampled global latent variable $\boldsymbol{z}$ as well as $\boldsymbol{A}$ and $\boldsymbol{X}$ as input, and outputs the probability distributions on $C$ for all nodes. Assume that the hidden representation of node $v$ in the $K^{th}$ layer is denoted as $\boldsymbol{e}_v^K$, and the initial latent representation of node $v$ is defined as the concatenation between $\boldsymbol{x}_v$ and $\boldsymbol{z}$, i.e., $\boldsymbol{e}_v^0 = \boldsymbol{x}_v||\boldsymbol{z}$. The predicted categorical distribution can be expressed as follows:

$$p_\theta(\boldsymbol{y}_v|\boldsymbol{A}, \boldsymbol{X}, \boldsymbol{z}) = Cat(\boldsymbol{y}_v|\boldsymbol{e}_v^K), \quad v \in V \tag{9}$$

**Model Training:** To optimize the object function in Eq. (6), we adopt Monte Carlo estimation to approximate the expectations $w.r.t$ $q_\phi(Y_U|\boldsymbol{A}, \boldsymbol{X}, Y_L)$ and $q_\phi(\boldsymbol{z}|\boldsymbol{A}, \boldsymbol{X}, Y)$. Specifically, we first sample $m$ instances of $Y_U$ from $q_\phi(Y_U|\boldsymbol{A}, \boldsymbol{X}, Y_L)$. After that, for each instance of $Y_U$, we further sample $n$ instances of $\boldsymbol{z}$ from $q_\phi(\boldsymbol{z}|\boldsymbol{A}, \boldsymbol{X}, Y)$. With these sampled instances, we could approximately estimate the object function $\mathcal{L}_{ELBO}(\theta, \phi)$. We leverage reparameterization to calculate the derivatives $w.r.t$ the parameters in $q_{net1}$, $q_{net2}$ and $p_{net}$. Since $\boldsymbol{z}$ is continuous and $q_\phi(\boldsymbol{z}|\boldsymbol{A}, \boldsymbol{X}, Y)$ takes on a Gaussian form, the reparameterization trick of variational auto-encoders [7] can be directly used here. While $Y_U$ is discrete, we adopt the Gumbel-Softmax reparametrization [23] for gradient backpropagation. As we mentioned above, $Y_L$ could be used as the supervised information to guide the parameter update of $q_{net1}$. Therefore, we additionally introduce a supervised object function $\mathcal{L}_s(\phi) = \log q_\phi(Y_L|\boldsymbol{A}, \boldsymbol{X})$. The overall objective function is given as follows:

$$\mathcal{L}(\theta, \phi) = \mathcal{L}_{ELBO}(\theta, \phi) + \mathcal{L}_s(\phi) \tag{10}$$

The model can be optimized effectively in an end-to-end manner and the optimal parameters are denoted by $\theta^*$ and $\phi^*$, i.e., $\theta^*, \phi^* = \arg\max_{\theta,\phi} \mathcal{L}(\theta, \phi)$.

**Model Inference:** After the above training, $p(Y_U|\boldsymbol{A}, \boldsymbol{X}, Y_L)$ can be seen as the expectation of $p_{\theta^*}(Y_U|\boldsymbol{A}, \boldsymbol{X}, \boldsymbol{z})$ $w.r.t.$ $q_{\phi^*}(\boldsymbol{z}|\boldsymbol{A}, \boldsymbol{X}, Y)$. We first sample $L$ instances of $Y_U$ from $q_{\phi^*}(Y_U|\boldsymbol{A}, \boldsymbol{X}, Y_L)$, and then for each sampled instance of $Y_U$, we sample a instance of $\boldsymbol{z}$. from $q_{\phi^*}(\boldsymbol{z}|\boldsymbol{A}, \boldsymbol{X}, Y)$. We use Monte Carlo estimation for approximate inference, formulated as follows:

$$p(Y_U|\boldsymbol{A}, \boldsymbol{X}, Y_L) \approx \frac{1}{L}\sum_{i=1}^{L} p_{\theta^*}(Y_U|\boldsymbol{A}, \boldsymbol{X}, \boldsymbol{z}_i) \tag{11}$$

This approximation can also be derived from Eq. (4) with the proof in the supplemental material.

### 3.3 Algorithm Complexity Analysis

Because $q_{net1}$ and $p_{net}$ share the similar message-passing model structure, the computational complexity of them is $\mathcal{O}(|E|)$, where $|E|$ represents the number of edges in the graph. The computational

complexity of $q_{net2}$ is $\mathcal{O}(N)$, where $N$ is the number of nodes. On this basis, during the training phase, the overall computational complexity is $\mathcal{O}(|E| + mN + mn|E|)$, where $m$ and $n$ are respectively the number of sampled instances of $Y_U$ and $z$. In our experiments, we find that one sample (*i.e.*, $m = n = 1$) could achieve comparable results with multiple samples. For efficiency, we only sample once for both $Y_U$ and $z$. During the inference phase, the calculation only involves $p_{net}$. Therefore, the overall computational complexity is $\mathcal{O}(L|E|)$, where $L$ is the number of sample instances from $p(z)$. We can see that the complexity is linear to the scale of the graph. The pseudo-code of the algorithm is provided in the supplemental material.

## 4 Experiments

In this section, we empirically evaluate the performance of GSNN on the task of semi-supervised node classification in different scenarios: (1) the standard experimental scenario with the validation set for early-stopping, (2) the scarce labeled nodes scenario (no validation set for early-stopping), and (3) the adversarial attack scenario. Note that we mainly consider the noise injected by adversarial attack methods, since they can incur obvious impact on the performance of many existing GNNs. [24, 25, 26, 27]. Our reproducible code is available at https://github.com/GSNN/GSNN.

### 4.1 Experimental Settings

**Datasets.** We conduct experiments on three commonly used benchmark datasets: Cora, Citeseer and Pubmed [25, 28], where nodes represent documents and edges represent citation relationships. Each node is associated with a bag-of-words feature vector and a ground-truth label. Detailed statistics for the three datasets are provided in the supplemental material. In different experiment scenarios, we will adopt different dataset setup (*e.g.*, dataset partition method) following the standard practice, which would be described when presenting experimental results in corresponding sections.

**Baselines.** When we evaluate the performance in the standard experimental settings and the scarce labeled nodes settings, we compared with six state-of-the-art models, three of which are GCN [3], GraphSAGE [4] and Graph Attention Networks (GAT) [5]. The other three adopt uncertainty modeling for graph-based semi-supervised learning. They are Bayesian Graph Convolutional Neural Networks (BGCN) [21], G$^3$NN [20] and Graph Gaussian Processes (GGP) [22] respectively. BGCN and G$^3$NN model the uncertainty of the graph structure, and GGP introduces the Gaussian processes to prevent from over-fitting. When we evaluate the performance in the adversarial attack settings, in addition to the above six baselines, we also compare with Robust Graph Convolutional Networks (RGCN) [29], which is a state-of-the-art method against adversarial attacks. More detailed description about the baselines are provided in the supplemental material.

**Our Model.** For the proposed GSNN framework, we could adopt different information aggregation mechanisms for $q_{net1}$ and $p_{net}$ to instantiate the models. In this paper, we implement two variants, whose aggregation mechanisms are consistent with GCN (*i.e.*, mean aggregation) [3] and GAT (*i.e.*, attention-based aggregation) [5] respectively. Note that other advanced information aggregation mechanisms can also be involved here to improve the performance. The two variants are termed as GSNN-M and GSNN-A.

**Parameter Settings.** For all baselines, we adopt the default parameter settings reported in corresponding papers. For our proposed two models (*i.e.*, GSNN-M and GSNN-A), in $q_{net1}$ and $p_{net}$, we employ two information aggregation layers, and other settings related to hidden layers are consistent with GCN [3] and GAT [5] respectively. For example, the number of hidden units for GSNN-M is set to 16 and that for GSNN-A is set to 64. Besides, GSNN-A also employs the multi-head attention mechanism in the first hidden layer with 8 attention heads. For both GSNN-M and GSNN-A, the dimension of the hidden variable $z$ is set to 16. In $q_{net2}$, we first employ a two-layer MLP to generate the representation $r_v$ for each node $v$, whose dimension is 16. After that, we summarize all representations into a vector and use two fully-connected networks to convert it into the mean and covariance matrix for the multivariate Gaussian distribution. As mentioned in Section 3.3, both the numbers of sampled instances of $Y_U$ and $z$ are set to 1 for efficiency purpose. We use the Adam optimizer [30] during training, with the learning rate as 0.01 and weight decay as $5 \times 10^{-4}$, and set the epoch number as 200. During the inference phase, the sampling number $L$ in Eq. (11) is set to 40.

In the experiments, we train our models and baselines for 50 times and record the mean classification accuracy and standard deviation.

## 4.2 Standard Experimental Scenario

In this section, we evaluate the performance of GSNN and baselines under the standard experimental scenario used in the work [3]. Specifically, in each dataset, 20 nodes per class are used for training, 1000 nodes are used for evaluation and another 500 nodes are used for validation and early-stopping.

The experimental results (mean and standard deviation) are summarized in Table 1. We can see that under the standard experimental scenario, BGCN, G$^3$NN and GGP do not show obvious advantages and perform even worse than the deterministic GNN-based models (*i.e.*, GCN, GAT and GraphSAGE) in many cases. The reason behind is that the validation set could help these GNN-based models find relatively good classification functions, which can prevent the model from overfitting to a large extent. Both BGCN and G$^3$NN attempt to model the uncertainty of the graph structure. However, the potential distributions of different graph data may vary greatly, which limits the performance of these two methods on some datasets (*e.g.*, Pubmed). GGP

Table 1: Average classification accuracy with standard deviation under the standard experimental scenario. The bold marker denotes the best performance on each dataset.

| Algorithm | Cora | Citeseer | Pubmed |
|---|---|---|---|
| GCN | $81.49 \pm 0.51$ | $70.34 \pm 0.70$ | $78.93 \pm 0.49$ |
| GAT | $83.01 \pm 0.40$ | $70.91 \pm 0.79$ | $78.57 \pm 0.75$ |
| GraphSAGE | $82.89 \pm 1.01$ | $70.08 \pm 0.78$ | $78.28 \pm 0.42$ |
| BGCN | $81.20 \pm 0.80$ | $72.20 \pm 0.60$ | $76.60 \pm 0.70$ |
| G$^3$NN | $82.90 \pm 0.30$ | $\mathbf{73.10} \pm 0.50$ | $77.60 \pm 0.70$ |
| GGP | $81.25 \pm 0.33$ | $67.18 \pm 0.66$ | $76.91 \pm 0.52$ |
| GSNN-M | $\mathbf{83.94} \pm 0.47$ | $72.20 \pm 0.45$ | $\mathbf{79.12} \pm 0.31$ |
| GSNN-A | $83.08 \pm 0.49$ | $71.66 \pm 0.23$ | $78.99 \pm 0.31$ |

adopts Gaussian processes to model the node classification task, of which the fitting capacity is not as good as neural networks that could effectively learn the node representations. Therefore, the performance of it is not ideal.

Compared with baselines, our models achieve comparable or better performance in standard experimental scenario. Note that GSNN-M and GSNN-A adopt the consistent aggregation mechanism with GCN and GAT respectively, while the results show that the two proposed models outperform GCN and GAT on all datasets, which demonstrates the effectiveness of modeling the uncertainty of the classification function.

## 4.3 Label-Scarce Scenario

In general, the labeled nodes are difficult or expensive to obtain. A more practical scenario is that we only have a very small proportion of labeled nodes for training and no additional labeled nodes for early-stopping. In this section, we evaluate the performance of GSNN and baselines when labeled nodes are scarce. Specifically, in each dataset, we randomly select a certain percentage of labeled nodes for training, and the rest of nodes are used for evaluation. Note that the number of labeled nodes in each class could be different under this dataset partition setting.

For Cora and Citeseer, we set the percentage of labeled nodes for training from 1% to 5%, while for Pubmed, we set the percentage from 0.1% to 0.5% because the total number of nodes in Pubmed is about an order of magnitude higher than the other two datasets. The experimental results are shown in Table 2. We observe that, compared with baselines, GSNN-M and GSNN-A achieve substantial performance gain, which demonstrates that modeling the uncertainty of the classification function could effectively alleviate the overfitting problem on the complex graph data. BGCN models the uncertainty of the graph structure, which improves the performance of the deterministic GNN-based models on Cora and Citesser to some extent. However, its performance cannot be generalized to Pubmed because of the difference of the potential graph structure for different datasets. Although G$^3$NN also models the distribution of the graph structure, the complex model structure make it easy to overfit without early-stopping. Therefore, modeling the distribution of the classification function provides more flexibility and better copes with the label-scarce scenario.

Table 2: Average classification accuracy under the label-scarce scenario with different ratios. The bold marker denotes the best performance on each dataset. Due to space constraints, we do not show the standard deviation here.

| Dataset | Ratio | GCN | GAT | GraphSAGE | BGCN | G$^3$NN | GGP | GSNN-M | GSNN-A |
|---|---|---|---|---|---|---|---|---|---|
| Cora | 5% | 79.87 | 80.61 | 79.08 | 81.69 | 78.05 | 75.80 | 82.21 | **82.49** |
|  | 4% | 79.35 | 80.22 | 78.89 | 80.85 | 75.07 | 72.41 | 82.11 | **82.44** |
|  | 3% | 78.42 | 79.33 | 78.52 | 80.51 | 62.74 | 68.91 | **82.69** | 81.66 |
|  | 2% | 76.73 | 77.96 | 76.82 | 77.98 | 47.11 | 56.30 | **81.05** | 79.94 |
|  | 1% | 66.58 | 70.09 | 68.18 | 71.23 | 32.95 | 46.71 | **71.76** | 71.62 |
| Citeseer | 5% | 70.55 | 69.41 | 68.40 | 71.45 | 70.72 | 65.11 | 71.24 | **71.89** |
|  | 4% | 69.11 | 68.33 | 67.13 | 70.37 | 70.41 | 64.61 | 69.74 | **71.10** |
|  | 3% | 68.26 | 67.11 | 65.54 | 70.18 | 65.04 | 58.49 | 70.26 | **70.88** |
|  | 2% | 67.01 | 67.37 | 66.41 | 68.31 | 56.16 | 53.18 | 68.47 | **70.24** |
|  | 1% | 60.08 | 61.39 | 61.25 | 63.25 | 30.28 | 49.57 | 62.21 | **64.91** |
| Pubmed | 0.5% | 82.18 | 80.01 | 81.32 | 78.25 | **82.73** | 78.97 | 82.17 | 80.70 |
|  | 0.4% | 80.85 | 79.09 | 79.82 | 76.32 | 81.53 | 75.86 | **81.70** | 79.92 |
|  | 0.3% | 79.98 | 77.95 | 79.51 | 75.62 | 79.80 | 75.25 | **80.69** | 79.10 |
|  | 0.2% | 76.33 | 77.01 | 77.54 | 73.01 | 76.59 | 59.28 | 78.12 | **78.89** |
|  | 0.1% | 69.21 | 70.99 | 71.42 | 67.92 | 42.46 | 55.92 | 72.23 | **73.17** |

## 4.4 Adversarial Attack Scenario

In this section, we employ three state-of-the-art global adversarial attack methods (*i.e.*, Meta-Train [25], Meta-Self [25] and min-max attack [26]), which aim at reducing the overall classification accuracy, to inject noise edges into the graph structure, and further evaluate the performance of GSNN and baselines in the presence of them. Detailed description for the three attack methods is provided in the supplemental material. The experimental settings about the adversarial attacks and dataset partition follow the work [25]. The attack budgets, *i.e.*, the ratio of perturbed edges to all clean edges, is set to 0.05. Without loss of generality, all three attack methods are performed based on the vanilla GCN [3], which means that they mainly affect the mean aggregation mechanism. For each poisoned graph, 10% of nodes are used for training and the rest of nodes are used for evaluation.

We conduct experiments on Cora. The experimental results are shown in Table 3. Here we add a robust GNN model (*i.e.*, RGCN [29]) as a baseline. We have the following meaningful observations: (1) Under the three attack methods, the performance of GCN reduces drastically because it serves as the surrogate model of the attacks. Meanwhile, they can transfer to other deterministic GNN-based models (*i.e.*, GraphSAGE and GAT). However, GSNN could effectively alleviate the impacts of attacks by modeling the uncertainty of the classification function. We can see that GSNN-M and GSNN-A significantly improve

Table 3: Average classification accuracy with standard deviation under the adversarial attack scenario on Cora. The bold marker denotes the best performance.

| Algorithm | Meta-Train | Meta-Self | min-max attack |
|---|---|---|---|
| GCN | $76.25 \pm 0.32$ | $75.17 \pm 0.38$ | $75.46 \pm 0.35$ |
| GAT | $81.35 \pm 0.62$ | $78.87 \pm 0.39$ | $81.37 \pm 0.47$ |
| GraphSAGE | $76.54 \pm 0.56$ | $75.69 \pm 0.48$ | $76.59 \pm 0.55$ |
| BGCN | $83.92 \pm 0.38$ | $81.29 \pm 0.41$ | $81.93 \pm 0.62$ |
| G$^3$NN | $83.14 \pm 0.36$ | $81.28 \pm 0.28$ | $80.68 \pm 0.47$ |
| GGP | $78.53 \pm 0.16$ | $77.14 \pm 0.11$ | $77.68 \pm 0.15$ |
| RGCN | $79.76 \pm 0.39$ | $77.85 \pm 0.45$ | $79.48 \pm 0.43$ |
| GSNN-M | $\mathbf{84.57} \pm 0.76$ | $80.13 \pm 0.66$ | $83.76 \pm 0.77$ |
| GSNN-A | $84.48 \pm 0.85$ | $\mathbf{82.15} \pm 0.70$ | $\mathbf{84.45} \pm 0.82$ |

the performance of GCN and GAT, and also outperform RGCN, which is a state-of-the-art method against the adversarial attacks. Note that although the attack methods mainly affect the mean aggregation mechanism, GSNN-M still maintains good performance. (2) BGCN and G$^3$NN could capture the underlying structure that exists in graph data. Therefore, they have capacity to improve the robustness against the adversarial attacks. Compared with them, GSNN does not need to modify the graph structure, which has more flexibility and achieves better or comparable performance.

# 5 Conclusion

In this paper, we propose a novel GSNN for semi-supervised learning on graph data, which aims to model the uncertainty of the classification function by simultaneously learning a family of functions. To model the distribution of the classification function, we introduce a learnable graph neural network coupled with a high-dimensional random latent vector, and further adopt the amortised variational inference to approximate the intractable joint posterior of the missing labels and the latent variable. Extensive experimental results show that GSNN outperforms the state-of-the-art baselines on different datasets. It shows great potential in label-scarce and adversarial attack scenarios. This paper focuses on the uncertainty of the GNN classification function. How to integrate more information, such as the label dependency and structure uncertainty, into the framework for inference is an interesting problem in the future.

## Acknowledgment

The authors would like to thank the anonymous reviewers for their valuable comments. This work was supported by the NSFC under Grant No. 11688101 and No. 61872360, the National Key Research and Development Program of China under Grant No. 2020YFE0200500, the ARC DECRA under Grant No. DE200100964, and the Youth Innovation Promotion Association CAS under Grant No. 2017210. Chuan Zhou, Jia Wu, Shirui Pan and Jilong Wang are corresponding authors.

## Broader Impact

Our work could bring the following positive impacts. (1) The proposed framework, which models the uncertainty of the classification function, provides a new idea for semi-supervised learning on graph data. (2) In practice, labeled nodes are generally scarce and expensive to obtain. GSNN could effectively alleviate the overfitting problem and improve the performance. (3) Noise could render deterministic GNN-based models vulnerable, while GSNN could alleviate the negative impacts of noise to a large extent. Many real-world applications, especially the risk-sensitive applications ($e.g.$, financial transaction), would benefit from it.

Similar with many other GNNs, one potential issue of our model is that it provides limited interpretation of its predictions. We advocate peer researchers to make a profound study on this to improve the interpretability of modern GNN architectures and make GNNs applicable in more risk-sensitive applications.

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
