[Supplementary Material]

# Graph Stochastic Neural Networks for Semi-supervised Learning: Supplemental Material

**Haibo Wang**[1, 2,*] **Chuan Zhou**[3, 4] **Xin Chen**[2,*] **Jia Wu**[5]**, Shirui Pan**[6]**, Jilong Wang**[2]
[1]Department of Computer Science and Technology, Tsinghua University, Beijing, China
[2]Institute for Network Sciences and Cyberspace, Tsinghua University, Beijing, China
[3]Academy of Mathematics and Systems Science, Chinese Academy of Sciences, Beijing, China
[4]School of Cyber Security, University of Chinese Academy of Sciences, Beijing, China
[5]Faculty of Science and Engineering, Macquarie University, Sydney, Australia
[6]Faculty of Information Technology, Monash University, Melbourne, Australia
{wang-hb15, cx18}@mails.tsinghua.edu.cn, zhouchuan@amss.ac.cn,
jia.wu@mq.edu.au, shirui.pan@monash.edu, wjl@cernet.edu.cn

The supplemental material includes the following contents. Note that the **symbols** and the **equation indexes** are consistent with those in the main paper.

- Theorem on Eq. (11) with the proof (Section 1).

- Pseudo-code of GSNN (Section 2).

- Detailed description for datasets (Section 3), baselines (Section 4), and adversarial attack methods used to inject noise edges into the graph structure (Section 5).

## 1    Theorem on Eq. (11) with the Proof

In the main paper, after training the model, we employ $p_{net}$ for inference with the expression shown in Eq. (11). Actually, Eq. (11) can also be derived from Eq. (4) rigorously.

**Theorem 5.1.** Let $\theta^*$ and $\phi^*$ denote the optimal parameters after model training. According to the definition in Eq. (4), the posterior distribution $p(Y_U|\boldsymbol{A}, \boldsymbol{X}, Y_L)$ can indeed be approximated as follows:

$$p(Y_U|\boldsymbol{A}, \boldsymbol{X}, Y_L) \approx \frac{1}{L} \sum_{i=1}^{L} p_{\theta^*}(Y_U|\boldsymbol{A}, \boldsymbol{X}, \boldsymbol{z}_i)$$

where $\{\boldsymbol{z}_1, \cdots, \boldsymbol{z}_L\}$ are sampled instances from the variational distribution $q_{\phi^*}(\boldsymbol{z}|\boldsymbol{A}, \boldsymbol{X}, Y)$.

**Proof.** Assume that $Y_U$ is a sampled instance from the variational distribution $q_{\phi^*}(Y_U|\boldsymbol{A}, \boldsymbol{X}, Y_L)$ (*i.e.*, $q_{net1}$), then we have:

$$q_{\phi^*}(\boldsymbol{z}|\boldsymbol{A}, \boldsymbol{X}, Y) = q_{\phi^*}(\boldsymbol{z}|\boldsymbol{A}, \boldsymbol{X}, Y_L, Y_U) = \frac{q_{\phi^*}(Y_U, \boldsymbol{z}|\boldsymbol{A}, \boldsymbol{X}, Y_L)}{q_{\phi^*}(Y_U|\boldsymbol{A}, \boldsymbol{X}, Y_L)}.$$

Since (1) $q_{\phi^*}(Y_U, \boldsymbol{z}|\boldsymbol{A}, \boldsymbol{X}, Y_L)$ is the variational distribution used to approximate the real joint posterior $p_{\theta^*}(Y_U, \boldsymbol{z}|\boldsymbol{A}, \boldsymbol{X}, Y_L)$, (2) $q_{\phi^*}(Y_U|\boldsymbol{A}, \boldsymbol{X}, Y_L)$ would be close to $p_{\theta^*}(Y_U|\boldsymbol{A}, \boldsymbol{X}, \boldsymbol{z})$ after model training, and (3) given $\boldsymbol{A}$, $\boldsymbol{X}$ and $\boldsymbol{z}$, the label information $Y_U$ and $Y_L$ are conditionally independent, we have:

$$q_{\phi^*}(\boldsymbol{z}|\boldsymbol{A}, \boldsymbol{X}, Y) \approx \frac{p_{\theta^*}(Y_U, \boldsymbol{z}|\boldsymbol{A}, \boldsymbol{X}, Y_L)}{p_{\theta^*}(Y_U|\boldsymbol{A}, \boldsymbol{X}, \boldsymbol{z})} = \frac{p_{\theta^*}(Y_U, \boldsymbol{z}|\boldsymbol{A}, \boldsymbol{X}, Y_L)}{p_{\theta^*}(Y_U|\boldsymbol{A}, \boldsymbol{X}, \boldsymbol{z}, Y_L)} = p_{\theta^*}(\boldsymbol{z}|\boldsymbol{A}, \boldsymbol{X}, Y_L).$$

---

Based on this approximation and the fact that distributions $p(\boldsymbol{z}|\boldsymbol{A}, \boldsymbol{X}, Y_L)$ and $p\big(\boldsymbol{y}_v | g_{\varphi_{Y_L}}(\boldsymbol{A}, \boldsymbol{X}; \boldsymbol{z})\big)$ can be modeled by parametric families of distributions $p_\theta(\boldsymbol{z}|\boldsymbol{A}, \boldsymbol{X}, Y_L)$ and $p_\theta(\boldsymbol{y}_v | \boldsymbol{A}, \boldsymbol{X}, \boldsymbol{z})$ respectively, it follows that

$$p(Y_U | \boldsymbol{A}, \boldsymbol{X}, Y_L) \triangleq \int p(\boldsymbol{z}|\boldsymbol{A}, \boldsymbol{X}, Y_L) p\big(Y_U | g_{\varphi_{Y_L}}(\boldsymbol{A}, \boldsymbol{X}; \boldsymbol{z})\big) \, d\boldsymbol{z}$$

$$= \int p_{\theta^*}(\boldsymbol{z}|\boldsymbol{A}, \boldsymbol{X}, Y_L) p_{\theta^*}(Y_U | \boldsymbol{A}, \boldsymbol{X}, \boldsymbol{z}) \, d\boldsymbol{z}$$

$$\approx \int q_{\phi^*}(\boldsymbol{z}|\boldsymbol{A}, \boldsymbol{X}, Y) p_{\theta^*}(Y_U | \boldsymbol{A}, \boldsymbol{X}, \boldsymbol{z}) \, d\boldsymbol{z}$$

Based on Monte Carlo estimation, $Y_U$ can therefore be approximately inferred as follows:

$$p(Y_U | \boldsymbol{A}, \boldsymbol{X}, Y_L) \approx \frac{1}{L} \sum_{i=1}^{L} p_{\theta^*}(Y_U | \boldsymbol{A}, \boldsymbol{X}, \boldsymbol{z}_i) \quad \text{where} \quad \boldsymbol{z}_i \sim q_{\phi^*}(\boldsymbol{z}|\boldsymbol{A}, \boldsymbol{X}, Y)$$

The proof of this theorem is completed. □

## 2 Pseudo-code of GSNN

The pseudo-code of GSNN is in Algorithm 1. Note that in our experiments, the numbers of sampled instances of $Y_U$ and $\boldsymbol{z}$ are both set to 1 (*i.e.*, $m = n = 1$) for efficiency purpose. The algorithm complexity of GSNN is linear to the scale of the graph.

---
**Algorithm 1** GSNN

---
**Input:** Graph $G$ with $A$, $X$ and $Y_L$, the numbers of sampled instances of $Y_U$ and $\boldsymbol{z}$: $m$ and $n$
**Output:** All parameters for $q_{net1}$, $q_{net2}$ and $p_{net}$
 1: Initialize all parameters for $q_{net1}$, $q_{net2}$ and $p_{net}$
 2: **while** $\mathcal{L}$ does not converge **do**
 3:     Calculate $q_\phi(\boldsymbol{y}_v | \boldsymbol{A}, \boldsymbol{X}, Y_L)$ and $h_v^{K-1}$ for each node $v$ using Eq. (7)
 4:     **for** i ← 1 to $m$ **do**
 5:         Sample $Y_U$ from the distribution $q_\phi(Y_U | \boldsymbol{A}, \boldsymbol{X}, Y_L)$
 6:         Calculate $q_\phi(\boldsymbol{z}|\boldsymbol{A}, \boldsymbol{X}, Y)$ based on the sampled $Y_U$ using Eq. (8)
 7:         **for** j ← 1 to $n$ **do**
 8:             Sample $\boldsymbol{z}$ from the distribution $q_\phi(\boldsymbol{z}|\boldsymbol{A}, \boldsymbol{X}, Y)$
 9:             Calculate $p_\theta(\boldsymbol{y}_v | \boldsymbol{A}, \boldsymbol{X}, \boldsymbol{z})$ based on the sampled $\boldsymbol{z}$ using Eq. (9)
10:         **end for**
11:     **end for**
12:     Calculate the objective function $\mathcal{L}(\theta, \phi)$ using Eq. (10)
13:     Update parameters of $q_{net1}$, $q_{net2}$ and $p_{net}$ by gradient descent.
14: **end while**

---

## 3 Detailed Statistics for Datasets

The detailed statistics of three datasets used in this paper are listed in Table 1. It summarizes the number of nodes, the number of edges, the dimension of features and the number of classes.

Table 1: Dataset statistics

| Dataset | #(Node) | #(Edge) | #(Feature) | #(Class) |
|---------|---------|---------|------------|----------|
| Cora | 2,708 | 5,429 | 1,433 | 7 |
| Citeseer | 2,110 | 3,757 | 3,703 | 6 |
| Pubmed | 19,717 | 44,324 | 500 | 3 |

# 4 Detailed Description of Baselines

In this paper, when evaluating the performance in the standard experimental scenario and in the label-scarce scenario, we compare with six state-of-the-art baselines used for graph-based semi-supervised learning. Three of them are deterministic GNN-based models, which are GCN [1], Graph Attention Networks (GAT) [2] and GraphSAGE [3] respectively. The other three adopt the uncertainty modeling, which are Bayesian Graph Convolutional Neural Networks (BGCN) [4], $G^3$NN [5] and Graph Gaussian Processes (GGP) [6] respectively. When evaluating the performance in the adversarial attack scenario, in addition to the above six baselines, we also compare with Robust Graph Convolutional Networks (RGCN) [7], which is a state-of-the-art method against adversarial attacks. The detailed description of these seven baselines are presented as follows:

- **GCN [1]:** It is one of the most classic GNN models, which defines the graph convolution in the spectral domain and uses the first-order approximation to reduce the number of parameters.
- **GAT [2]:** It defines the graph convolution in the spatial domain, which introduces the attention mechanism to assign different weights to different neighbor nodes when aggregating the neighbor information.
- **GraphSAGE [3]:** It defines the graph convolution in the spatial domain, which samples a fixed-size set of neighbors for information aggregation.
- **BGCN [4]:** It treats the observed graph as a realization from a parametric family of random graphs and performs the inference for the joint posterior of random graph parameters and node labels.
- **$G^3$NN [5]:** It is a generative framework that models the joint distribution of node features, labels, and graph structure, which also treats the graph as a random variable.
- **GGP [6]:** It introduces Gaussian processes to model the semi-supervised learning problem on graphs and employs the scalable variational inference algorithm to perform the posterior inference.
- **RGCN [7]:** It adopts Gaussian distributions as the hidden representations of nodes to absorb the effects of adversarial attacks into the variances and employs a variance-based attention mechanism to remedy the propagation of adversarial attacks.

# 5 Detailed Description of Adversarial Attack Methods

In Section 4.4 of the main paper, we evaluate the performance of GSNN and baselines in the presence of three state-of-the-art adversarial attack methods, of which the detailed description is as follows:

- **Meta-Train [8]:** This method treats the adjacent matrix as a hyperparameter to be optimized and further computes the meta-gradient of the attack loss $w.r.t.$ it. It greedily modifies one edge in each step based on the maximum gradient until the attack budget is reached. The attack loss is calculated via nodes in the training set.
- **Meta-Self [8]:** This method is a variant of Meta-Train. It calculates the attack loss based on nodes in the test set, where the labels are predicted by a trained surrogate model.
- **min-max attack [9]:** This method models the attack as a *min-max* optimization problem and recovers a binary solution based on the discrete sampling after the optimization.