[Reviews · NeurIPS 2020]

Review 1

Summary and Contributions: I do not think the rebuttal addresses my concerns on any of the 3 issues: (1) mathematical mistakes in inference (2) extremely high cost (3) wrong experiment setting. For (1), the correct formulation is p(Y_U|A,X,z) = \mathbb{E}_{q(z|A,X,Y)} p(Y_U|A,X,z), then you need to sample from q(z|A,X,Y) and do Monte Carlo instead of prior, this is a serious mistake. For (2), I keep my opinion that a 4000% increase of computation time does not compensate for a 4% relative improvement. For (3), the authors claim "the total number of labeled nodes in the case of 0.5% under the label-scarce scenario is still more than that under the standard experimental scenario", then why do the authors claim that this is low-label setting?? This setting has more labels than standard scenario... ------------------------------------------------------ The work targets semi-supervised node classification tasks and aims to design a robust graph neural network using a latent variable model. The idea is to model the joint distribution with latent variables and use mean-field variational inference for approximations.

Strengths: The general formulation is concrete. The authors nicely organize the paper with formulation of the latent variable model and then describe in detail how they instantiate each module. There is also extensive analysis of the complexity of the algorithm. I find the figure helpful and make the paper clearer. The mathematical notations are considered consistent throughout the paper. The authors conduct extensive experiments on multiple setups including normal node classification, low-label regime and adversarial defence against malicious attack. The proposed model also shows strong empirical performance over previous baselines in all three setups with comparable standard deviation. This is pretty impressive considering GSNN contains more stochasticity than the baselines.

Weaknesses: This paper combines latent variable models with GNNs, it’s not novel enough and there are many previous works with similar ideas in graph generation. The difference is that the formulation of this paper is more like a conditional generative model and targets at node classification tasks. Based on the implementation of the method, I think the model is similar to RGCN in some aspects. Undoubtedly, there are differences that the model does not directly learn a Gaussian representation but instead samples from a Gaussian latent variable and concatenates it with the features of the node. However, both aim to inject some noise and in essence decrease the information between the representation and the original node feature so that the model only captures the key attributes and thus making the model more robust than vanilla GNNs. One concern is about the inference procedure. Why does the model directly sample from prior instead of the posterior? I find the statement “Since the posterior would be close to the prior after model training, so we directly sample from prior” (line 181-184) very sketchy. I believe the learned latent space (if trained well) should be structured in the sense that if we train a VAE on MNIST, then the posterior of the same digits will be close while the posterior of different digits will be scattered across the latent space. It’s not like after training, the posteriors will look exactly the same as prior. If it is the case, then unfortunately it means that the mutual information between z and the node will be close to 0 and should be useless. Another issue is that I feel the comparison with baselines in the experiments is not fair since the proposed model needs to sample $z$ for $L$ times, and $L=40$ in the experiments. It means that the model is at least **40** times slower than vanilla GNNs and can be worse with the additional overhead. It will be super helpful if the authors can show the results with different $L$, e.g. 1, 5, 20, 40. Another minor point is that GSNN also has more parameters, but I think it should be fine if the authors can show GSNN has comparable parameters with GAT. Also in the adversarial attack setting, where the graph is attacked on the structure with adding/deleting edges, I am not fully convinced that the model can alleviate this issue simply by noise injection in feature. I feel a nicer way that can possibly achieve more performance boost is to also model the structure $A$ using latent variable models. I list some additional points below. - There lacks an explanation as to why the authors did not use $Y_L$ as input for qnet1. - The paper does not have a clear description on how they (randomly) selected nodes for training in the label-scarce scenario. For example, is it completely random or did you select a fixed number of nodes for each class? I also did not find the details in the appendix. - In low data regime, another important previous work is PPNP [1], which also shows strong performance over standard GNNs. I suggest that the authors compare their method with PPNP. - Why does the model have higher performance in the scarce-label setting than the clean setting? As listed in Table 1 and 2, GSNN-M/A has higher accuracy in 0.5% Pubmed than in the original Pubmed. Also same for adversarial attack, it seems that the model can achieve a higher score on the perturbed graph than on the pristine graph. This is rather counterintuitive. [1] Predict then Propagate: Graph Neural Networks meet Personalized PageRank

Correctness: I think the method is correct in high-level. I am concerned about the inference steps, which I also mentioned in the weakness section.

Clarity: Overall I think the paper is easy to follow. Some suggestions and typos include: - line 91: moset -> most - line 291: stat-of-the-art -> state-of-the-art - Some notations in figure 1 need to be in bold so that they are consistent with the main body. - $N$ is the same as $|V|$ in Sec. 3.3, there is no need to further introduce $N$ as a new notation to represent the number of nodes.

Relation to Prior Work: The paper has extensive discussions but also has room for improvement with discussion and comparison between RGCN and GSNN.

Reproducibility: Yes

Additional Feedback:


Review 2

Summary and Contributions: Most existing models learn a deterministic classification function, which lack sufficient flexibility to explore better choices in the presence of noisy observations, scarce labeled nodes, and noisy graph structure. This paper proposes a novel framework named Graph Stochastic Neural Networks (GSNN), to model the uncertainty of the classification function by simultaneously learning a family of stochastic functions.

Strengths: (1) The paper propose a stochastic framework for GNN, which models the uncertainty of the classification function by simultaneously learning a family of stochastic functions. (2) Comparison against several baselines. (3) The paper is well written

Weaknesses: (1) The paper uses only three small datasets (core, citeseer, pubmed) with a relatively high autocorrelation, this makes it difficult to see how the proposed method would generalize to other graphs with more noise and low autocorrelation. (2) The two proposed methods (GSNN-M, GSNN-A) seems to provide similar results, it's unclear which one is better for which graphs, i.e., for different graphs with different noise in attributes and structure. It'd be good to explore these with synthetic experiments, and add a discussion on them.

Correctness: I think the claims are correct.

Clarity: The paper is mostly well written, however it should be checked for typos, for example: - Line 91 "Moset existing GNN ..." -> "Most exising GNN ..."

Relation to Prior Work: Related work is discussed.

Reproducibility: Yes

Additional Feedback: ===After author response and discussion with other reviewers=== I thank the Authors for responding to my comments in an acceptable way.


Review 3

Summary and Contributions: This paper attempts to address an unresolved but meaningful problem. To improve the inflexibility of GNNs in the face of imperfect observed data, the paper proposes a novel framework GSNN to model the uncertainty of classification function by simultaneously learning a family of functions. GSNN treats the classification function as a stochastic function, and uses a learnable graph neural network parameterized by a high-dimensional latent variable to model its distribution. To infer the missing labels by classification function with uncertainty, GSNN wisely adopts variational inference technology to approximate the intractable joint posterior for missing labels and the latent variable. The extensive experimental results show that GSNN achieves substantial performance gain in different scenarios, such as the label-scarce scenario and adversarial attack scenario.

Strengths: Significance and novelty: Most GNN-based models learn a deterministic classification function, which makes them lack sufficient flexibility to cope with kinds of imperfect observed data, such as scarce labels or deliberate noise in the graph structure. To solve the problems, this paper proposes to model the uncertainty of the classification function and simultaneously learn a family of functions, which is well-motivated. The idea is novel and different from previous works, which provide a new perspective for the graph-based semi-supervised learning problem. Soundness of the claims: The authors skillfully formalize the problem of modeling the uncertainty of classification function. They treat the classification function to be learned as a stochastic function and further combine GNN models and a high-dimensional latent variable to model its distribution. The variational inference technology makes the missing labels become inferable. The overall solutions of this paper, including theoretical analysis, practical model design and experiment evaluation are technically sound. Extensive experimental results also show significant performance gain compared with stat-of-the-art baselines, which further demonstrates the effectiveness of the proposed method. Relevance: This paper has the potential to attract wide attention at NeurIPS 2020.

Weaknesses: 1.Theoretically, this work fills the gap between the deterministic classification function and the stochastic classification function. A toy motivation example in the introduction is encouraged. By doing this, the proposed GSNN work will reach a broader audience. 2.Some typos in the paper.

Correctness: The basic theory and the practical model design are clear and correct. The overall logic and derivation are rigorous. It is praiseworthy for the authors to provide a proof to claim the consistency between the practical approximation inference (i.e., Eq. (11)) and theoretical definition (Eq. (4)), which is meaningful to demonstrate the completeness and correctness of the proposed method. The overall experiments are well designed and the evaluation results could support most claims made by the authors. It is better to describe the parameter settings of baselines and the proposed model in Section 4.1 with more details.

Clarity: The paper is well organized and easy to follow. The formalization is clear and the notations are defined rigorously and keep consistent throughout the paper.

Relation to Prior Work: The paper carefully claims the differences between the proposed model and two types of related works, including deterministic graph neural networks and methods using uncertainty modeling. To make the audiences understand the problem more clearly, the baselines in this paper, especially the second type of methods, need to be introduced in more details.

Reproducibility: Yes

Additional Feedback: (1)    Some typos need to be revised. For example, “Moset” in line 91 should be “Most”. (2)    The parameter settings of baselines and the proposed model should be described with more details. (3)    It would be appreciated if the authors could provide a toy motivation example to make the idea more intuitive and understandable.

[Author Response · NeurIPS 2020]

**To Reviewer #2 and #3:** Thanks for your positive comments. We will involve your suggestions in the revised version.

**To Reviewer #1:** Thank you for the comments. You have some misunderstandings on this paper.

C1: It is similar with (conditional) graph generation. R1: **Our basic idea is different from the existing graph gener-**
**ative models**. The latter ones aim to **model the distribution of the observed data**, while our method (GSNN) treats
the classification function as a stochastic one and attempts to **model the distribution of the stochastic classification**
**function**, which is a new idea for semi-supervised learning on graph data.

C2: It is similar to RGCN. R2: **Our model is different from RGCN**. RGCN models the node representations
as Gaussian distributions, while **GSNN models the uncertainty of the classification function by introducing the**
**random latent variable $\mathfrak{z}$**. Specifically, for GSNN, **concatenating the sampled $z$ and the node feature $x$ is an**
**implementation way to introduce randomness into the classification function, whose objective is not to model**
**the node feature**. Note that a sampled instance of $z$ could instantiate a classification function, which acts on all nodes
in the graph. In other words, **feature vectors for all nodes share the same instance of $z$**.

C3: Why sample from prior instead of the posterior? R3: We agree that sampling from the posterior is a usual practice.
Actually, our model is also designed following this practice as shown in Fig. 1 of the main paper. In the training
phase, based on the reparameterization trick, we could sample $z$ from the posterior and optimize the model parameters.
However, in the testing phase, to infer the missing labels, we still need to sample multiple instances of $Y_U$ from
$q_\phi(Y_U|\boldsymbol{A}, \boldsymbol{X}, Y_L)$ before sampling $z$ from the posterior $q_{\phi^*}$, which involves $q_{net1}$, $q_{net2}$ and $p_{net}$, making it inflexible.
We notice that the second item of the ELBO objective function in Eq. (6) is the opposite of the KL-divergence between
the posterior $q_{\phi^*}(\boldsymbol{z}|\boldsymbol{A}, \boldsymbol{X}, Y)$ and the prior $p(\boldsymbol{z})$, which can be seen as a regularizer to encourage the posterior to be
close to the prior. Under this observation, we here adopt a simpler method, directly sampling $z$ from the prior $p(\boldsymbol{z})$, to
construct multiple classification functions. This method only involves $p_{net}$ and is more flexible. The test results tell us
these two kinds of methods are comparable.

C4: Concern about the mutual information between $z$ and the node. R4: The meaning of the introduced random latent
variable $\mathfrak{z}$ is different from that in VAE or CVAE. As shown in Eq. (2) and lines 101 to 106 of the main paper, $\mathfrak{z}$ **is**
**not to distinguish different types of nodes, but to introduce randomness into the classification function** $g_\varphi$ (*i.e.*,
$p_{net}$). $\mathfrak{z}$ can be viewed as a parameter of $p_{net}$. Given a sampled $z$ from $\mathfrak{z}$, $p_{net}$ will specify an instance of classification
function to classify all nodes in the graph. In other words, an instance of $z$ corresponds to a classification function, not
a node in the graph. It is not the case of VAE on MNIST.

C5: The comparison is not fair since GSNN needs to sample $z$ for $L = 40$ times and it is 40 times slower than GNNs.

R5: **Since our proposed model is a Bayesian one, it is a standard practice to sample**
**multiple samples of $z$ for estimating the proposed GSNN model.** As to the efficiency,
during the training phase, the number of the sampled instance of $\boldsymbol{z}$ is set to 1 as mentioned
in lines 236-237, which would not increase the time complexity compared with vanilla
GNNs. While in the inference phase, as shown in Eq. (11), we need sample $L$ instances
of $\boldsymbol{z}$ to achieve the Monte Carlo estimation. However, the inference only requires $L$
feedforwards of $p_{net}$, which runs very fast. We plot Fig. A to show the change of the
performance $w.r.t.$ $L$. We can observe that the average classification accuracy keeps
high and stable when $L > 10$ for all three datasets.

Figure A: Accuracy $w.r.t.$ $L$.

C6: Why the model can alleviate structure attacks by noise injection in feature? Modeling the structure $A$ is a nicer
way. R6: (1) **The performance gain benefits from modeling the randomness of the classification function, rather**
**than the node feature. Please refer to R2.** (2) We agree that modeling the graph structure is another potential way to
deal with adversarial attacks, which, however, is not the main target of this paper. Besides, we included BGCN and
G$^3$NN, which model the distribution of the graph structure, as our baselines. The experimental results show that the
proposed GSNN achieves better performance.

C7: Why not use $Y_L$ as input for $q_{net1}$. R7: As shown in lines 148-149 and Fig. 1 in the main paper, $Y_L$ is used as the
supervision information for training $q_{net1}$ in the Eq. (10), which serves as the input of $q_{net1}$ indirectly.

C8: How to select labeled nodes in the label-scarce scenario. R8: We select a certain percentage of nodes completely
randomly. We will include it in our revised version.

C9: Suggest PPNP as a baseline. R9: Thank you for your suggestion. We will include PPNP as a baseline.

C10: Why performance is higher in the label-scarce and adversarial attack settings than the original graphs? R10: The
dataset partition method under the standard experimental scenario is different from these under the label-scarce and
adversarial attack scenarios. For example, on Pubmed, the total number of labeled nodes in the case of 0.5% under the
label-scarce scenario is still more than that under the standard experimental scenario.

[Meta-Review · NeurIPS 2020]

The majority of the reviewers are in consensus that this is solid research and should be accepted.